# HIV and antiretroviral treatment knowledge gaps and psychosocial burden among persons living with HIV in Lima, Peru

Rafaella Navarro[1,2⊙], Jose Luis Paredes[1,2⊙] *, Juan Echevarria[1,2,3], Elsa González-Lagos[1,2], Ana Graña[2,3], Fernando Mejía[1,2,3], Larissa Otero[1,2]

1 Facultad de Medicina, Universidad Peruana Cayetano Heredia, Lima, Perú, 2 Instituto de Medicina Tropical Alexander von Humboldt, Universidad Peruana Cayetano Heredia, Lima, Perú, 3 HIV Program, Hospital Cayetano Heredia, Lima, Perú

⊙ These authors contributed equally to this work.
* jose.luis.paredes.s@upch.pe

**Data Availability Statement:** All relevant data are within the paper and its Supporting Information files.

## Abstract

This study aims to describe knowledge on HIV and antiretroviral (ARV) treatment and psychosocial factors among people living with HIV (PLWH) in Lima, Perú, to explore characteristics associated to this knowledge, and determine its impact on sustained viral suppression. A cross-sectional survey was conducted among 171 PLWH at the largest referral health care center in Lima. The psychosocial factors measured were depression, risk of alcoholism, use of illegal drugs and disclosure. A participant had "poor knowledge" when less than 80% of replies were correct. Sustained viral suppression was defined as two consecutive viral loads under 50 copies/mL. A total of 49% and 43% had poor HIV and ARV knowledge respectively; 48% of the study population screened positive for depression and 27% reported feeling unsupported by the person they disclosed to. The largest gaps in HIV and ARV knowledge were among 98 (57%) that did not recognize that HIV increased the risk of cancer and among 57 (33%) participants that did not disagree with the statement that taking a double dose of ARV if they missed one. Moderate depression was significantly associated to poor HIV and ARV knowledge. Non-disclosure and being on ARVs for less than 6 months were associated with not achieving sustained viral suppression. Our findings highlight important HIV and ARV knowledge gaps of PLWH and a high burden of psychosocial problems, especially of depression, among PLWH in Lima, Peru. Increasing knowledge and addressing depression and disclosure could improve care of PLWH.

## Introduction

HIV treatment expansion has resulted in a 51% decline in AIDS-related deaths globally from 2004 to 2017 [1]. However, HIV remains the second cause of death from an infectious disease with 1.8 million new HIV infections in 2017 [1]. The UNAIDS Fast Track strategy proposes ending the AIDS epidemic by 2030 by increasing coverage of tested, treated and virally

**Funding:** This study was funded by the Program for Advanced Research Capacities for AIDS in Peru (PARACAS) [grant number D43TW00976301] from the Fogarty International Center at the U.S. National Institutes of Health (NIH). The funder had no role in study design, data collection, data analysis, decision to publish or preparation of the manuscript.

**Competing interests:** The authors have declared that no competing interests exist.

suppressed, to reach 95% of all persons living with HIV, decreasing new infections to 200 000 among adults and zero discrimination [2].

The key for HIV control is sustained viral suppression by ensuring adherence to antiretrovirals (ARV) as detectable viral loads are associated with increased transmission, morbimortality and drug resistance. Health related knowledge and health literacy increase adherence to ARV and empower persons to participate in their own care [3,4]. Yet, significant gaps on knowledge on HIV transmission and treatment among people living with HIV (PLWH) have been described [5–8]. Knowledge is only one of the determinants of adherence to antiretrovirals [9,10]. Poor mental health including depression, substance abuse and heavy alcohol consumption have also been recognized as barriers to care in PLWH and are associated with low adherence, unsuppressed viremia disease progression, and mortality among PLWH [11–14].

In Lima, Peru, although the level of HIV related knowledge among PLWH has not been quantified in this context, some studies shed a light on health information and outcomes among PLWH. In a study in Lima, 77% of PLWH said that they understood all information given by the doctors [15] and a qualitative study in Piura, Peru concluded that PLWH on ARV have important misconceptions on HIV transmission and treatment and maintain sexual behaviors that can facilitate HIV transmission [16].

We conducted this study to quantify HIV and ARV knowledge among PLWH attending the largest HIV referral center in Lima, and to analyze the association of demographic characteristics and psychosocial factors -specifically depression, disclosure and substance abuse–to the level of knowledge. Finally, we analyzed the impact of HIV and ARV knowledge on viral suppression controlling for psychosocial factors that could also affect viral suppression.

## Materials and methods

### Ethical considerations

The Institutional Review Boards of Universidad Peruana Cayetano Heredia and Hospital Cayetano Heredia approved the study protocol. Linking of the study database and the hospital databases was done with a unique numeric ID. All study researchers were certified in responsible conduct of research. All participants in this study provided a written consent to participate to the study, after explanation of the risks and benefits of participation of this study.

### Study design, setting and population

By 2016, there were 66 907 PLWH notified in Peru [17]. We conducted a cross-sectional study at the HIV program of a tertiary hospital in Lima, with a catchment area of 2,682,608 inhabitants and which provides care to the largest number of PLWH in Peru [18]. Participants were considered eligible if they were PLWH over 18 years old, registered in the hospital HIV program and able to provide written consent.

### Study procedures

Eligible participants were invited to participate while waiting for routine blood sampling for viral load measurements, between November 2016 and July 2017. Those consenting were requested to answer the self-administered paper-based questionnaire, which included 18 multiple-choice questions: eleven on HIV knowledge and seven on ARV knowledge. HIV and ARV knowledge were analyzed separately since we hypothesized that they could be influenced differently by psychosocial factors, their association with viral suppression could be independent, and two previous study have studied both separately [6,8]. Questions to measure knowledge were developed based on two published surveys [6,8], and in consultation with infectious

diseases clinicians and nurses providing HIV care. To determine clarity, any potential discomfort or alternative responses, the questionnaire was tested with six PLWH. The questionnaire also included 14 questions on psychosocial factors (the standardized Mental Health Inventory-5 (MHI5) scale [19] for depression, the CAGE questionnaire for risk of alcoholism [20], illegal drug consumption, disclosure and perception of support by the disclosed ones) as potential determinants of knowledge and of not achieving sustained viral suppression. Six other potential determinants (age, sex, educational and marital status, time from enrollment in the HIV program and time since ARV initiation) and viral load measurements were extracted from the hospital records.

## Data management and analysis

Data was entered in an Access database and analyzed using Stata v15. We calculated percentages for categorical variables and median and interquartile ranges for continuous variables of participant´s demographics, psychosocial factors and HIV and ARV knowledge. Both knowledge questionnaires were scored according to importance and implications of the knowledge addressed by infectious diseases clinicians and nurses. Each correct answer in the HIV knowledge section received 0.25 points (maximum score = 2.75). Four questions on ARV knowledge received one point and three received 0.25 points for correct answers (maximum score = 4.75). We defined "good knowledge" when 80% or more of the maximum score was obtained, and "poor knowledge" when less than 80%.

Sustained viral suppression was defined as having two consecutive viral loads with less than 50 cop/mL in a period of 12 months, we used the viral load measured on the enrollment day and the most proximate within 6 months after or before the survey.

To study the variables associated with three outcomes: poor HIV knowledge, poor ARV knowledge and not achieving sustained viral suppression we used Poisson regression to calculate prevalence ratios in the bivariate and multivariate analysis. For the multivariate analysis we included variables with a p value <0.2 in the bivariate analysis and we used backwards elimination: variables with the weakest association to the dependent variable were taken off one to one until a significant difference with the previous model was found by likelihood ratio test.

To study the determinants of poor HIV and ARV knowledge, we included factors that could be associated with poor knowledge: sociodemographic factors (sex, age, marital status and educational status), psychosocial factors (depression, disclosure, perception of support from the person to whom they had disclosed, risk of alcoholism, use of illegal drugs), time from enrollment to the HIV program and the study interview and time on ARVs. Finally, to study the role of HIV and ARV knowledge on not achieving sustained viral suppression we included PLWH who were on ARV and who had the two viral load measurements available and controlled for the factors mentioned above.

## Results

### Study population and participant's characteristics

Of 255 eligible participants, 205 were enrolled, 171 completed more than 50% of the knowledge section of the survey and thus were included in the analysis of HIV and ARV knowledge. Of the 171 participants, 152 (88.9%) had two viral load measurements available and were included in the analysis for sustained viral suppression (Fig 1). Of the participants included, 121/171 (70.8%) were male, the median age was 36 (IQR 28–44). A total of 47 (27.5%), were married or cohabiting, 5 (2.9%) divorced, 108 (63.2%) single and 11 (6.4%) widowers. Eighty-five (49.7%) participants completed high school, 12 (7.0%) primary school and 74 (43.3%)

Eligible participants invited to participate (n= 255)
  -    PLWH older than 18 enrolled in care in the HIV program from
       HCH
  -    Able to read and understand the questionnaire

  -    Declined participation (n= 50, 19.6%)
       o   Lack of time (n=32, 64%)
       o   No reason given (n=10, 20%)
       o   Could not read well (n=8, 16%)
  -    34 participants (13.3%) filled less than 50%
       of survey and were excluded from the
       analysis

Enrolled in the study and included in the analysis of HIV and ARV knowledge
(n= 171)
  -    152 participants (88.9%) were on ARV, had two viral load
       measurements available and were included in the analysis for sustained
       viral suppression.

**Fig 1. Study population, people living with HIV in a referral center in Lima, Peru, 2016–2017.**

university/technical studies. The median time between participants' enrollment in the HIV program and the study interview was 4.0 years (IQR, 1.6–7.6) and the median time between ARV start date and the interview was 3.2 years (IQR 1.1–6.4). Ten (5.8%) participants were not on ARV.

## Psychosocial factors

Of 171 PLWH, 154 (90.1%) reported disclosure of their HIV diagnosis, 6 (3.5%) did not disclose and 11 (6.4%) did not reply. Of the 154 patients who disclosed their HIV status, 112 (72.7%) felt supported at least by someone they disclosed to and 42 (27.3%) by no one. According to the MHI5 questionnaire, 72/171 (42.1%) were not depressed, 29/171 (17.0%) were mild depressed, 32/171 (18.7%) were moderately depressed, 21/171 (12.3%) were severely depressed and 17/171 (9.9%) did not reply. On the CAGE questionnaire, 123/171 (71.9%) PLWH were not at risk of alcoholism, 20 (11.7%) were at risk of alcoholism and 28/171 (16.4%) did not answer. Regarding use of illicit drugs 143/171 (83.6%) replied that they had never used drugs, 11/171 (6.4%) PLWH reported ongoing drug use, 12/171 (7.0%) reported using it in the past and 5/171 (2.9%) did not answer.

## HIV and ARV knowledge

The median score on general HIV knowledge among 171 participants was 2.25 (IQR 1.75–2.5) and on ARV knowledge was 4 (IQR 3–4.75). Eighty-three participants (48.5%) had poor HIV

**Table 1. Knowledge on HIV and ARV, among people living with HIV in a referral center in Lima, Peru, 2016–2017 (N = 171).**

| | Correct | Incorrect | Does not know | Do not answer |
|---|---|---|---|---|
| **Knowledge on HIV** | | | | |
| **Is HIV an illness that can be cured or controlled?** | 150 (87.7) | 8 (4.7) | 11 (6.4) | 2 (1.2) |
| **A PLWH can live the same number of years as a person not infected?** | 126 (73.7) | 12 (7.0) | 27 (15.8) | 6 (3.5) |
| **I must use a condom in any sexual relation with a person without HIV** | 156 (91.2) | 7 (4.1) | 7 (4.1) | 1 (0.6) |
| **I am at risk of another infection if I have sex with a PLWH without using a condom** | 136 (79.5) | 21 (12.3) | 11 (6.4) | 3 (1.8) |
| **I have to use condom if I have sex with a PLWH.** | 138 (80.7) | 15 (8.8) | 15 (8.8) | 3 (1.8) |
| **The use of microbicides during sex avoids HIV transmission** | 81 (47.4) | 28 (16.4) | 57 (33.3) | 5 (2.9) |
| **If I use condoms correctly I will have safe sex** | 149 (87.1) | 11 (6.4) | 8 (4.7) | 3 (1.8) |
| **HIV infection places me at higher risk of** | | | | |
| **Cancer** | 74 (43.3) | 31 (18.1) | 63 (36.3) | 4 (2.3) |
| **Sexually transmitted infections** | 121 (70.8) | 15 (8.8) | 33 (19.3) | 2 (1.2) |
| **Diarrhea** | 133 (77.8) | 14 (8.2) | 20 (11.7) | 4 (2.3) |
| **Dental problems** | 83 (48.5) | 30 (17.5) | 53 (31.0) | 5 (2.9) |
| **Knowledge on ARV** | | | | |
| **The following substances interfere with ARV** | | | | |
| **Large amounts of alcohol** | 117 (68.4) | 23 (13.5) | 18 (10.5 | 13 (7.6) |
| **Marijuana** | 113 (66.1) | 20 (11.7) | 27 (15.8) | 11 (6.4) |
| **Cocaine** | 127 (74.3) | 15 (8.8) | 17 (9.9) | 12 (7.0) |
| **While I am on ARV, I do not transmit HIV** | 136 (79.5) | 19 (11.1) | 14 (8.2) | 2 (1.2) |
| **If I forget to take my ARV, I can make up for it by taking a double dose the following day** | 114 (66.7) | 30 (17.5) | 24 (14.0) | 3 (1.8) |
| **If I am feeling ok, I can discontinue ARV** | 147 (86.0) | 8 (4.7) | 14 (8.2) | 2 (1.2) |
| **It is important to discontinue ARV for a few days to rest the body** | 141 (82.5) | 14 (8.2) | 14 (8.2) | 2 (1.2) |

knowledge and 74 (43.3%) poor ARV knowledge. The largest gap in HIV knowledge was among 98 (57.3%) that did not reply correctly that HIV increased the risk of cancer and among 90 (52.6%) that did not disagree with "The use of microbicides during sex avoids HIV transmission" (Table 1). The largest gap in ARV knowledge was among 30 (17.5%) who agreed with "If I forget to take my ARV, I can make up for it by taking a double dose the following day" and among 23 (13.5%) who believed that large amounts of alcohol does not interfere with ARV (Table 1).

## Determinants of HIV and ARV knowledge

Tables 2 and 3 show the bivariate and multivariate analysis of the demographic and psychosocial characteristics associated with HIV and ARV knowledge, respectively. Age 44 years old or more, moderate depression, receiving ARV <0.5 years and receiving ARV for 0.5–1 years were significantly associated with poor HIV knowledge in the bivariate and multivariate analysis. Single participants were less likely to have poor HIV knowledge. Moderate depression and not replying to the survey on depression, were associated to poor ARV knowledge in the bivariate and multivariate analysis. In the bivariate analysis, we found a possible interaction effect between sex and use of illegal drugs and between depression and age, but had insufficient power to test in the multivariate analysis.

## Determinants of not achieving sustained viral suppression

Among 152 participants who were on ARV and who had two viral loads available, 62 (40.8%) did not achieve sustained viral suppression. The bivariate and multivariate analysis of the association of HIV and ARV knowledge and potential determinants of not achieving sustained

**Table 2. Bivariate and multivariate analysis of determinants of knowledge on HIV among people living with HIV in a referral center in Lima, Peru, 2016–2017 (N = 171).**

| | Bivariate analysis | | | | Multivariate analysis | |
|---|---|---|---|---|---|---|
| | Good HIV knowledge[a] | Poor HIV knowledge[a] | PR Crude (95% CI) | P value | Adjusted PR (95% CI) | P value |
| **Age group** | | | | | | |
| 18–27 | 25 (61.0) | 16 (39.0) | 1 | | 1 | |
| 28–36 | 25 (55.6) | 20 (44.4) | 1.1 (0.7–1.9) | 0.61 | 1.1 (0.7–1.9) | 0.68 |
| 37–43 | 24 (57.1) | 18 (42.9) | 1.1 (0.7–1.8) | 0.72 | 1.3 (0.7–2.2) | 0.43 |
| ≥44 | 14 (32.6) | 29 (67.4) | 1.7 (1.1–2.7) | 0.01 | 1.6 (1.1–2.6) | 0.034 |
| **Sex** | | | | | | |
| Male | 68 (56.2) | 53 (43.8) | 1 | | 1 | |
| Female | 20 (40.0) | 30 (60.0) | 1.4 (1.01–1.9) | 0.04 | 1.0 (0.7–1.4) | 0.93 |
| **Marital Status** | | | | | | |
| Married or Cohabiting | 15 (31.9) | 32 (68.1) | 1 | | 1 | |
| Divorced | 3 (60.0) | 2 (40.0) | 0.6 (0.2–1.8) | 0.34 | 0.5 (0.2–1.5) | 0.21 |
| Single | 65 (60.2) | 43 (39.8) | 0.6 (0.4–0.8) | <0.01 | 0.7 (0.5–0.9) | 0.02 |
| Widowers | 5 (45.5) | 6 (54.5) | 0.8 (0.5–1.4) | 0.45 | 0.9 (0.4–1.8) | 0.75 |
| **Educational status** | | | | | | |
| Primary School | 6 (50.0) | 6 (50.0) | 0.9 (0.5–1.7) | 0.80 | [b] | [b] |
| High School | 39 (45.9) | 46 (54.1) | 1 | | | |
| Superior Education | 43 (58.1) | 31 (41.9) | 0.8 (0.6–1.1) | 0.13 | | |
| **Mental health scale by MHI-5** | | | | | | |
| Not Depressed | 43 (59.7) | 29 (40.3) | 1 | | 1 | |
| Mild Depressed | 17 (58.6) | 12 (41.4) | 1.0 (0.6–1.7) | 0.91 | 1.0 (0.6–1.6) | 0.98 |
| Moderate Depressed | 9 (28.1) | 23 (71.9) | 1.8 (1.2–2.5) | <0.01 | 1.7 (1.2–2.5) | <0.01 |
| Severe Depressed | 10 (47.6) | 11 (52.4) | 1.3 (0.8–2.1) | 0.30 | 1.3 (0.8–2.1) | 0.26 |
| Did not answer | 9 (52.9) | 8 (47.1) | 1.2 (0.7–2.1) | 0.60 | 1.3 (0.7–2.3) | 0.43 |
| **Alcoholism screening risk by CAGE** | | | | | | |
| No Abuse | 62 (50.4) | 61 (49.6) | 1 | | [b] | [b] |
| Risk Abuse | 9 (45.0) | 11 (55.0) | 1.1 (0.7–1.7) | 0.64 | | |
| Did not answer | 17 (60.7) | 11 (39.3) | 0.8 (0.5–1.3) | 0.36 | | |
| **Use of illegal drugs** | | | | | | |
| Used in the past | 3 (25.0) | 9 (75.0) | 1.6 (1.1–2.3) | 0.02 | [b] | [b] |
| Sometimes | 6 (54.5) | 5 (45.5) | 1.0 (0.5–1.9) | 0.33 | | |
| Never used | 75 (52.4) | 68 (47.6) | 1 | | | |
| Did not answer | 4 (80.0) | 1 (20.0) | 0.4 (0.1–2.5) | 0.89 | | |
| **Disclosure of HIV diagnosis** | | | | | | |
| At least to someone | 78 (50.6) | 76 (49.4) | 1 | | [b] | [b] |
| No one | 4 (66.7) | 2 (33.3) | 0.7 (0.2–2.1) | 0.50 | | |
| Did not answer | 6 (54.5) | 5 (45.5) | 0.9 (0.5–1.8) | 0.81 | | |
| **Self-perception of support among those disclosing** | | | | | | |
| Supported by someone | 56 (50.0) | 56 (50.0) | 1 | | [b] | [b] |
| Supported by no one | 22 (52.4) | 20 (47.6) | 1.0 (0.7–1.4) | 0.80 | | |
| Did not disclosed | 4 (66.7) | 2 (33.3) | 0.7 (0.2–2.1) | 0.49 | | |
| Did not answer | 6 (54.6) | 5 (45.5) | 0.9 (0.5–1.8) | 0.78 | | |
| **Time from enrollment to the HIV program and the study interview (in years)** | | | | | | |
| 0–0.5 | 7 (50.0) | 7 (50.0) | 0.9 (0.2–3.9) | 0.53 | [b] | [b] |
| 0.51–1.5 | 11 (44.0) | 14 (56.0) | 1.0 (0.3–3.1) | 0.30 | | |

*(Continued)*

**Table 2.** (Continued)

| | Bivariate analysis | | | | Multivariate analysis | |
|---|---|---|---|---|---|---|
| | Good HIV knowledge[a] | Poor HIV knowledge[a] | PR Crude (95% CI) | P value | Adjusted PR (95% CI) | P value |
| 1.51–4.0 | 23 (47.9) | 25 (52.1) | 1.2 (0.5–3.0) | 0.23 | | |
| 4.01–8.6 | 25 (53.2) | 22 (46.8) | 1.0 (0.4–2.8) | 0.53 | | |
| >8.6 | 22 (59.5) | 15 (40.5) | 1 | | | |
| **Time from ARV and the study interview (in years)** | | | | | | |
| No ARV | 6 (60.0) | 4 (40.0) | 1.4 (0.6–3.1) | 0.93 | 1.4 (0.6–3.1) | 0.43 |
| <0.5 | 5 (33.3) | 10 (66.7) | 1.9 (1.1–3.4) | 0.08 | 1.9 (1.1–3.4) | 0.03 |
| 0.51–1.5 | 11 (39.3) | 17 (60.7) | 2.1 (1.2–3.6) | 0.12 | 2.1 (1.2–3.6) | <0.01 |
| 1.51–3.5 | 26 (54.2) | 22 (45.8) | 1.5 (0.9–2.8) | 0.55 | 1.5 (0.9–2.8) | 0.14 |
| 3.51–6.5 | 24 (54.5) | 20 (45.5) | 1.4 (0.8–2.4) | 0.58 | 1.4 (0.8–2.4) | 0.29 |
| >6.51 | 16 (61.5) | 10 (38.5) | 1 | | 1 | |

MHI-5: Mental Health Inventory-5 ARVs: Antiretroviral.

[a]Each question received a 0.25 score for a correct answer. Incorrect answers, does not know or does not answer gave 0 points. The maximum possible score was 2.75.
We defined "good knowledge" when ≥80% of the maximum score was obtained, and "poor knowledge" if < 80%.
[b]omitted because of collinearity.

viral suppression are shown in Table 4. In the bivariate analyses, moderate depression was associated with not achieving sustained viral suppression, this association did not remain in the multivariate analysis. Non-disclosure and being on ARV less than 0.5 years were associated with not achieving sustained viral suppression in the bivariate and multivariate analysis.

## Discussion

Among PLWHA attending a referral center in Lima, up to a quarter had gaps in key knowledge on HIV and ARV and 41% had not achieved sustained viral suppression. We also found a high burden of psychosocial problems: 48% of study participants screened positive for any grade of depression and 27% reported feeling unsupported by the person they disclosed to. Moderate depression was associated to poor HIV and ARV knowledge. Non-disclosure was associated with not achieving sustained viral suppression.

In general, HIV and ARV knowledge was high in our study; however, a quarter of the study population had knowledge gaps that could impact HIV care and transmission: 9% believed that they should not use a condom in sexual contact with other PLWH and 12% said that sex with another PLWH does not pose a risk of other infections. Considering that serosorting (choosing a sexual partner that is also living with HIV) is common among PLWH, our result highlights knowledge gaps that might affect this practice [7]. It is important to note, this study was implemented before the launch of the UNAIDS undetectable = untransmittable campaign [21]. The question with the lowest proportion of participants replying correctly was on the knowledge of the higher risk of cancer posed by HIV infection. In Nigeria, poor knowledge on AIDS defining cancers among PLWH has also been described and few participants had undergone cancer screening and they attributed this to the lack of knowledge of its benefits [22]. Further research should test strategies to increase knowledge about their risk of cancer and to increase compliance to screening practices to reduce cancer mortality among PLWH.

We found gaps on ARV knowledge, such as 18% of the population that thought they could make up a missed dose of ARV by taking a double dose the next day. In our study, 8% considered the statement "It is important to discontinue ARV for a few days to rest the body" correct,

**Table 3. Bivariate and multivariate analysis of determinants of antiretroviral (ARV) knowledge, among people living with HIV in a referral center in Lima, Peru, 2016–2017 (N = 171).**

| | Bivariate analysis | | | | Multivariate analysis | |
| --- | --- | --- | --- | --- | --- | --- |
| | Good ARV knowledge[a] | Poor ARV knowledge[a] | PR Crude (95% CI) | P value | Adjusted PR (95% CI) | P value |
| **Age groups** | | | | | | |
| 18–27 | 29 (70.7) | 12 (29.3) | 1 | | 1 | |
| 28–36 | 27 (60.0) | 18 (40.0) | 1.4 (0.8–2.5) | 0.31 | 1.3 (0.8–2.3) | 0.29 |
| 37–43 | 22 (52.4) | 20 (47.6) | 1.6 (0.9–2.9) | 0.10 | 1.6 (0.9–2.8) | 0.09 |
| ≥44 | 19 (44.2) | 24 (55.8) | 1.9 (1.1–3.3) | 0.02 | 1.6 (0.9–2.6) | 0.09 |
| **Sex** | | | | | | |
| Male | 73 (60.3) | 48 (39.7) | 1 | | [b] | [b] |
| Female | 24 (48.0) | 26 (52.0) | 1.3 (0.9–1.9) | 0.13 | | |
| **Marital Status** | | | | | | |
| Married or Cohabiting | 22 (46.8) | 25 (53.2) | 1 | | 1 | |
| Divorced | 2 (40.0) | 3 (60.0) | 1.1 (0.5–2.4) | 0.76 | 1.1 (0.5–2.3) | 0.87 |
| Single | 70 (64.8) | 38 (35.2) | 0.7 (0.5–1.0) | 0.03 | 0.8 (0.5–1.1) | 0.14 |
| Widowers | 3 (27.3) | 8 (72.7) | 1.4 (0.9–2.1) | 0.18 | 1.2 (0.8–1.9) | 0.42 |
| **Educational status** | | | | | | |
| Primary School | 5 (41.2) | 7 (58.3) | 1.1 (0.7–1.9) | 0.72 | [b] | [b] |
| High School | 40 (47.1) | 45 (52.9) | 1 | | | |
| Technical/university education | 52 (70.3) | 22 (29.7) | 0.6 (0.4–0.8) | <0.01 | | |
| **Mental health scale by MHI-5** | | | | | | |
| Not Depressed | 50 (69.4) | 22 (30.6) | 1 | | 1 | |
| Mild Depressed | 20 (69.0) | 9 (31.0) | 1.0 (0.5–1.9) | 0.96 | 0.9 (0.5–1.8) | 0.79 |
| Moderate Depressed | 8 (25.75) | 24 (75.0) | 2.5 (1.6–3.6) | <0.01 | 2.0 (1.4–3.0) | <0.01 |
| Severe Depressed | 13 (61.9) | 8 (38.1) | 1.2 (0.7–2.4) | 0.51 | 1.1 (0.6–2.1) | 0.69 |
| Did not answer | 6 (35.3) | 11 (64.7) | 2.1 (1.3–3.5) | <0.01 | 1.8 (1.1–2.9) | 0.02 |
| **Alcoholism screening risk by CAGE** | | | | | | |
| No Abuse | 70 (56.9) | 53 (43.1) | 1 | | [b] | [b] |
| Risk Abuse | 10 (50.0) | 10 (50.0) | 1.2 (0.7–1.8) | 0.55 | | |
| Did not answer | 17 (60.7) | 11 (39.3) | 0.9 (0.6–1.5) | 0.72 | | |
| **Use of illegal drugs** | | | | | | |
| Used in the past | 5 (41.7) | 7 (58.3) | 1.4 (0.8–2.3) | 0.73 | [b] | [b] |
| Sometime | 7 (63.6) | 4 (36.4) | 0.9 (0.4–1.9) | 0.21 | | |
| Never used | 83 (58.0) | 60 (42.0) | 1 | | | |
| Did not answer | 2 (40.0) | 3 (60.0) | 1.4 (0.7–3.0) | 0.35 | | |
| **Disclosure of HIV diagnosis** | | | | | | |
| At least to someone | 88 (57.1) | 66 (42.9) | 1 | | [b] | [b] |
| No one | 3 (50.0) | 3 (50.0) | 1.2 (0.5–2.7) | 0.71 | | |
| Did not answer | 6 (54.5) | 5 (45.5) | 1.1 (0.5–2.1) | 0.86 | | |
| **Self-perception of support among those disclosed** | | | | | | |
| Supported by someone | 63 (56.3) | 49 (43.8) | 1 | | [b] | [b] |
| Supported by no one | 25 (59.5) | 17 (40.5) | 0.9 (0.6–1.4) | 0.72 | | |
| Did not disclosed | 3 (50.0) | 3 (50.0) | 1.1 (0.5–2.6) | 0.75 | | |
| Did not answer | 6 (54.5) | 5 (45.4) | 1.0 (0.5–2.1) | 0.91 | | |
| **Time from ARV and the study interview (in years)** | | | | | | |
| No ARV | 5 (50.0) | 5 (50.0) | 1.3 (0.6–3.0) | 0.52 | [b] | [b] |
| 0–0.5 | 6 (40.0) | 9 (60.0) | 1.6 (0.8–3.0) | 0.17 | | |
| 0.51–1.5 | 18 (64.3) | 10 (35.7) | 0.9 (0.5–1.9) | 0.84 | | |

(*Continued*)

**Table 3.** (Continued)

| | Bivariate analysis | | | | Multivariate analysis | |
|---|---|---|---|---|---|---|
| | Good ARV knowledge[a] | Poor ARV knowledge[a] | PR Crude (95% CI) | P value | Adjusted PR (95% CI) | P value |
| 1.51–3.5 | 29 (60.4) | 19 (39.6) | 1.0 (0.6–2.0) | 0.93 | | |
| 3.51–6.5 | 23 (52.3) | 21 (47.7) | 1.2 (0.7–2.3) | 0.46 | | |
| >6.51 | 16 (61.5) | 10 (38.5) | 1 | | | |
| Time from enrollment to the HIV program and the study interview (in years) | | | | | | |
| 0–0.5 | 4 (28.6) | 10 (71.4) | 1.9 (1.1–3.2) | 0.02 | [b] | [b] |
| 0.51–1.5 | 17 (68.0) | 8 (32.0) | 0.8 (0.4–1.7) | 0.64 | | |
| 1.51–4.0 | 27 (56.3) | 21 (43.8) | 1.2 (0.7–2.0) | 0.59 | | |
| 4.01–8.6 | 26 (55.3) | 21 (44.7) | 1.2 (0.7–2.0) | 0.53 | | |
| >8.61 | 23 (62.2) | 14 (37.8) | 1 | | | |

MHI-5: Mental Health Inventory-5 ARVs: Antiretrovirals.

[a] Four of the seven questions on ARV knowledge received one point and three received 0.25 points, for a correct answer. Incorrect answers, does not know or does not answer gave 0 points. The maximum score for this section was 4.75.

[b] omitted because of collinearity.

as compared to 34% of low income latino PLWH in Los Angeles in 2003 [6]; these studies were conducted 15 years apart and even though our populations had a better knowledge, the presence of this beliefs can impact HIV spread and treatment. In our study 86% of participants disagreed with the statement "If I am feeling ok, I can discontinue ARV" which is similar to a 2012 study in Nigeria, where 92% disagreed with the statement: "You should take ARVs only when you feel sick" [8]. It is worrying that some patients, albeit few, do not know the importance of consistent adherence to ARVs.

Depression may affect our ability to understand a disease [23,24]. Depression is a barrier to care and is associated with lower adherence to ARV [25,26]. High frequency of depression among PLWH have been reported in Lima: 68% among HIV-positive impoverished women [27] and 48% among patients with HIV and TB [28]; both studies used the Hopkins Symptoms checklist. In a study using the WHOQOL-BREF questionnaire depressed HIV patients had a significantly lower quality of life than their no depressed counterparts [29]. In Chile, PLWH with moderate-severe depressive symptoms had three times higher risk of non-adherence compared to patients with mild to no depressive symptoms [30]. The prevalence of depression in 2012 among adults in the general population of Lima, Peru was estimated 17.2% measured with the Mini Mental State Examination [31]. Our study highlights a high frequency of depression (48%) and its role as a determinant of HIV and ARV knowledge in PLWH at a referral center in Lima, which reveals an urgent need to develop interventions to address depression among them. A meta-analysis suggested short-term improvements in depression and a significant reduction in viral load with cognitive behavioral therapy, while another meta-analysis suggested that pharmacological interventions were more effective [32,33]. Finally, during the study Peruvian guidelines for first line ARV included efavirenz, which has been associated with severe depression, suicidal ideation and nonfatal suicide attempts [34–36].

The proportion of PLWH that had disclosed their HIV status in our study was high and similar to that reported in a global study among 2035 PLWH where 96% had disclosed and the pooled proportion for Latin America was 92% [37]. In our study non-disclosure was found to be associated with not achieving sustained viral suppression. The relationship between disclosure, adherence and viral suppression remains poorly understood. In a meta-analysis, eleven

**Table 4. Bivariate and multivariate analysis of determinants of not achieving sustained viral suppression among people living with HIV in a referral center in Lima, Peru, 2016–2017 (N = 152).**

| | Bivariate analysis | | | | Multivariate analysis | |
|---|---|---|---|---|---|---|
| | Sustained viral suppression | Not sustained viral suppression | PR Crude (95% CI) | P value | Adjusted PR (95% CI) | P value |
| **Knowledge on HIV** | | | | | | |
| Good | 45(60.0) | 30(40.0) | 1 | | - | - |
| Poor | 45(58.4) | 32(41.6) | 1.0 (0.7–1.5) | 0.85 | | |
| **ARVs Knowledge** | | | | | | |
| Good | 55(65.5) | 29(34.5) | 1 | | 1 | |
| Poor | 35(51.5) | 33(48.5) | 1.4 (0.9–2.1) | 0.08 | 1.5 (0.8–1.7) | 0.47 |
| **Age Group** | | | | | | |
| 18–27 | 17(51.5) | 16(48.5) | 1 | | - | - |
| 28–36 | 24(58.5) | 17(41.5) | 0.9 (0.5–1.4) | 0.55 | | |
| 37–43 | 26(66.7) | 13(33.3) | 0.7 (0.4–1.2) | 0.20 | | |
| > = 44 | 23(59.0) | 16(41.0) | 0.8 (0.5–1.4) | 0.52 | | |
| **Sex** | | | | | | |
| Male | 65(60.2) | 43(39.8) | 1 | | - | - |
| Female | 25(56.8) | 19(43.2) | 1.1 (0.7–1.6) | 0.70 | | |
| **Marital Status** | | | | | | |
| Married or Cohabiting | 27(61.4) | 17(38.6) | 1 | | - | - |
| Divorced | 3(60.0) | 2(40.0) | 1.0 (0.3–3.2) | 0.95 | | |
| Single | 52(56.5) | 40(43.5) | 1.1 (0.7–1.7) | 0.60 | | |
| Widowers | 8(72.7) | 3(27.3) | 0.7 (0.3–2.0) | 0.51 | | |
| **Educational status** | | | | | | |
| Primary School | 6(75.0) | 2(25.0) | 0.6 (0.2–1.9) | 0.34 | - | - |
| High School | 42(54.6) | 35(45.5) | 1 | | | |
| University | 42(62.7) | 25(37.3) | 0.8 (0.6–1.2) | 0.33 | | |
| **Mental health scale by MHI-5** | | | | | | |
| Not Depressed | 41(66.1) | 21(33.9) | 1 | | 1 | |
| Mild Depressed | 17(68.0) | 8(32.0) | 0.9 (0.4–1.8) | 0.87 | 0.9 (0.5–1.6) | 0.67 |
| Moderate Depressed | 13(43.3) | 17(56.7) | 1.7 (1.0–2.7) | 0.03 | 1.5 (0.9–2.4) | 0.12 |
| Severe Depressed | 9(45.0) | 11(55.0) | 1.6 (0.9–2.8) | 0.07 | 1.5 (0.9–2.5) | 0.12 |
| Did not answer | 10(66.7) | 5(33.3) | 1.0 (0.4–2.2) | 0.69 | 1.0 (0.4–2.2) | 0.96 |
| **Alcoholism screening risk by CAGE** | | | | | | |
| No Abuse | 64(59.8) | 43(40.2) | 1 | | - | - |
| Risk of Abuse | 8(40.0) | 12(60.0) | 1.5 (1.0–2.3) | 0.06 | | |
| Did not answer | 18(72.0) | 7(28.0) | 0.7 (0.4–1.4) | 0.29 | | |
| **Use of illegal drugs** | | | | | | |
| Used in the past | 6(50.0) | 6(50.0) | 1.0 (0.4–2.2) | 0.43 | - | - |
| Sometimes | 6(60.0) | 4(40.0) | 1.2 (0.7–2.2) | 0.51 | | |
| Never used | 74(59.2) | 51(40.8) | 1 | | | |
| Did not answer | 4(80.0) | 1(20.0) | 0.5 (0.1–2.9) | 0.96 | | |
| **Disclosure of HIV diagnosis** | | | | | | |
| At least to someone | 82(59.9) | 55(40.2) | 1 | | 1 | - |
| No one | 1(20.0) | 4(80.0) | 2.0 (1.2–3.2) | 0.01 | 1.8 (1.2–2.9) | <0.01 |
| Did not answer | 7(70.0) | 3(30.0) | 0.7 (0.3–2.0) | 0.56 | 0.8 (0.3–2.0) | 0.61 |
| **Self-perception of support among those disclosing** | | | | | | |
| Supported by someone | 55(56.7) | 42(43.3) | 1 | | - | - |

*(Continued)*

**Table 4.**  (Continued)

| | Bivariate analysis | | | | Multivariate analysis | |
|---|---|---|---|---|---|---|
| | Sustained viral suppression | Not sustained viral suppression | PR Crude (95% CI) | P value | Adjusted PR (95% CI) | P value |
| Supported by no one | 27(67.5) | 13 (32.5) | 0.8 (0.5–1.2) | 0.26 | | |
| Did not disclosed | 1(20.0) | 4(80.0) | 1.8 (1.1–3.0) | 0.02 | | |
| Did not answer | 7(70.0) | 3(30.0) | 0.7 (0.3–1.8) | 0.46 | | |
| Time from ARV and the study interview (in years) | | | | | | |
| <0.5 | 3 (20.0) | 12 (80.0) | 2.9 (1.5–5.6) | <0.01 | 2.6 (1.3–5.2) | <0.01 |
| 0.51–1.5 | 17(68.0) | 8(32.0) | 1.1 (0.5–2.7) | 0.76 | 1.1 (0.5–2.5) | 0.87 |
| 1.51–3.5 | 27(61.4) | 17(38.6) | 1.4 (0.7–2.9) | 0.39 | 1.3 (0.6–2.7) | 0.50 |
| 3.51–6.5 | 25 (58.1) | 18 (41.9) | 1.5 (0.7–3.1) | 0.28 | 1.5 (0.7–3.0) | 0.28 |
| >6.51 | 18 (72.0) | 7 (28.0) | 1 | | 1 | |

MHI-5: Mental Health Inventory-5 ARVs: Antiretrovirals.

of seventeen studies reported a positive finding between disclosure and ARV adherence [38]. In a qualitative study in Peru the fear of disclosure was recognized by PLWH as a barrier to adherence to ARV [39]. Disclosure supposedly increases social support, which allows PLWH to cope with health and drug use [40]. However due to HIV-related stigma, disclosure might affect negatively ARV adherence [38] and it has been proposed that rather than promoting disclosure, programs should create supportive environments for PLWH, which might have more of an impact on adherence [41]. In our study 27% reported feeling unsupported by the person they disclosed. Disclosure to supportive persons should be encouraged among PLWH and studies should focus on understanding its association with retention in care.

The 12% risk or high risk of alcoholism found in our study is lower compared to studies among PLWH using the same questionnaire: in Boston, 42% PLWH had a risk or high risk of alcoholism, however, the study selected patients with higher pretest risk [42]. Another study, found that 43% of HIV-infected Peruvian MSM and transgender women had alcohol use disorders and 5% had alcohol dependence using the AUDIT test; both were inversely related to optimal ARV adherence [43]. The CAGE and AUDIT scores have the same sensitivity [44]. In the general population older than 15 years in Peru, 22% report excessive alcohol consumption [45]. Alcohol consumption and substance abuse have been associated to low adherence to ARV and to unsuppressed viremia [9,46].

Forty-one percent did not reach sustained viral suppression. This percentage is lower than the national estimates in which 63% have not achieved viral suppression, defined as <1000 cop/mL [47]. Our study included persons attending a referral hospital and thus we did not include patients not retained in care. Two studies conducted in the same hospital than this study found similar percentages of unsuppressed viremia: one found that 24% had a detectable viral load (defined as a single viral load above 1000 copies/mL including patients with at least 24 weeks on ARV) [48], the other study found that 40% of PLWH had a single detectable viral load (defined as having an viral load above 200 copies/mL within the first year of enrolment in HIV programs) [49]. The two factors associated with not achieving sustained viral suppression were non-disclosure and being on ARVs less than 0.5 years. The association between being on ARVs less than 0.5 years and not achieving sustained viral suppression was expected, since most patients achieve viral suppression after six months of ARVs [50,51].

Our study has several limitations and strengths. The questionnaire to measure HIV and ARV knowledge was not formally validated. However, it was based on validated surveys and

questionnaires, developed in consultation with HIV experts and we performed a pilot study to address any misunderstanding. By enrolling participants at the hospital waiting room for their viral load measures we might have overestimated knowledge; since our population did not include PLWH not retained in care. It is estimated that in the study hospital, between 53% and 57% of PLWH are retained in care after one year of entering the HIV Program [52]. Therefore, our results cannot be extrapolated to all the PLWH population but to that retained in care in Lima hospitals. PLWH that accepted to participate in our study may have been more educated than those not participating, since the former might have felt more comfortable replying to a questionnaire on knowledge. However, 43% study participants had completed high school which is comparable with the proportion in Lima (48%) [53]. The strength of our study is that, in addition to measuring knowledge, we surveyed several psychosocial issues that can be key for adherence and viral suppression. Our results allow the generation of several hypotheses related to knowledge, adherence and psychosocial factors among PLWH that can be tested in future studies. For example, if early identification and treatment of depression and encouraging disclosure may increase adherence.

Peruvian guidelines on HIV management recommend that doctors, nurses and a psychologist, should guide PLWH in their care and provide knowledge on HIV treatment and transmission [35]. In 2017, after the study conclusion, a psychiatrist was appointed to the HIV program. We suggest continuing prioritizing access to psychological evaluation and support especially in the identification and management of depression. We also suggest that the burden of other mental disorders and their impact on retention in care among PLWH should be quantified.

This study highlights important gaps in HIV and ARV knowledge and a high burden of psychosocial problems, especially of depression, among PLHW attending HIV care at a referral hospital. Addressing psychosocial factors that may be modified by health interventions, such as depression and disclosure could improve the quality of life among PLWH in Lima and improve HIV specific outcomes. Finally, our findings suggest, that it is necessary to develop interventions to address HIV and ARV knowledge gaps that could impact HIV transmission and treatment success in PLWH in Lima. Larger studies addressing the determinants of HIV and ARV knowledge and factors associated to HIV could test hypotheses resulting from our findings.

## Supporting information

**S1 Table. Score used to quantify knowledge on HIV and on ARV.**
(DOCX)

**S1 File. Study database.**
(DTA)

## Acknowledgments

We acknowledge the collaboration of Dyana Guardia, RN; Suzette Olivares, RN and Ms. Sandra Bejarano.

## Author Contributions

**Conceptualization:** Rafaella Navarro, Jose Luis Paredes, Juan Echevarria, Elsa González-Lagos, Ana Graña, Fernando Mejía, Larissa Otero.

**Data curation:** Rafaella Navarro, Jose Luis Paredes, Larissa Otero.

**Formal analysis:** Rafaella Navarro, Jose Luis Paredes, Elsa González-Lagos, Larissa Otero.

**Funding acquisition:** Rafaella Navarro, Jose Luis Paredes, Elsa González-Lagos, Larissa Otero.

**Investigation:** Rafaella Navarro, Jose Luis Paredes, Juan Echevarria, Elsa González-Lagos, Ana Graña, Fernando Mejía, Larissa Otero.

**Methodology:** Rafaella Navarro, Jose Luis Paredes, Juan Echevarria, Elsa González-Lagos, Ana Graña, Fernando Mejía, Larissa Otero.

**Project administration:** Rafaella Navarro, Jose Luis Paredes, Ana Graña, Larissa Otero.

**Resources:** Rafaella Navarro, Jose Luis Paredes, Larissa Otero.

**Software:** Rafaella Navarro, Jose Luis Paredes, Larissa Otero.

**Supervision:** Juan Echevarria, Elsa González-Lagos, Fernando Mejía, Larissa Otero.

**Validation:** Rafaella Navarro, Jose Luis Paredes, Juan Echevarria, Elsa González-Lagos, Ana Graña, Fernando Mejía, Larissa Otero.

**Visualization:** Jose Luis Paredes, Larissa Otero.

**Writing – original draft:** Rafaella Navarro, Jose Luis Paredes, Larissa Otero.

**Writing – review & editing:** Juan Echevarria, Elsa González-Lagos, Ana Graña, Fernando Mejía, Larissa Otero.

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
