## [Decision Letter · Decision Letter 0]

20 Apr 2021

PONE-D-21-01332

HIV and antiretroviral treatment knowledge gaps and psychosocial burden among persons living with HIV in Lima, Peru

PLOS ONE

Dear Dr. Paredes,

Thank you for submitting your manuscript to PLOS ONE. After careful consideration, we feel that it has merit but does not fully meet PLOS ONE’s publication criteria as it currently stands. Therefore, we invite you to submit a revised version of the manuscript that addresses the points raised during the review process.

The manuscript has been evaluated by two reviewers, and their comments are available below.

The reviewers have raised a number of major concerns. Specifically, they request significant improvements to the reporting of methodological aspects of the study, as well as further elaboration on the rationale for the study. They also request clarity to some aspects of the results reported.

Could you please carefully revise the manuscript to address all comments raised?

We look forward to receiving your revised manuscript.

Kind regards,

Avanti Dey, PhD

Staff Editor

PLOS ONE

Journal Requirements:

Reviewers' comments:

Reviewer's Responses to Questions

**Comments to the Author**

1. Is the manuscript technically sound, and do the data support the conclusions?

Reviewer #1: Yes

Reviewer #2: Yes

2. Has the statistical analysis been performed appropriately and rigorously? 

Reviewer #1: Yes

Reviewer #2: Yes

3. Have the authors made all data underlying the findings in their manuscript fully available?

Reviewer #1: No

Reviewer #2: No

4. Is the manuscript presented in an intelligible fashion and written in standard English?

Reviewer #1: Yes

Reviewer #2: No

5. Review Comments to the Author

Reviewer #1: A well-written paper that provides information about the levels of knowledge of HIV and ARV treatment as well as the prevalence of psycho-social factors among consenting PLWH waiting for viral load measurement at the waiting room of a tertiary healthcare institution in Lima, Peru. To further improve the paper quality the following are suggested,

1. The title should be revised to include the fact that the participants were consenting PLWH at the waiting room of the hospital for viral load measurement. This is necessary as there is no sample size estimation and the representativeness as well as the generalisability of the findings are questionable.

2. The rationale for this work needs to be further clarified. The authors mentioned that the level of HIV related knowledge among PLWH has not been quantified, but cited studies that indicated that PLWH understood all information given by their doctors. Besides, qualitative studies on HIV transmission reveals that misconception on HIV transmission exists. The authors should indicate the gap in knowledge that this study was out to bridge.

3. The authors recognised the importance of adherence to ARV as a key to sustained viral suppression in HIV control in their introduction yet did not measure this important variable by any means from the participants in the study. It would have been good to know the role of adherence as a potential confounder or middle variable in the association between viral suppression and knowledge or psychosocial factors.

4. The aspect of the paper which sought to analyse and proved information about the "impact of HIV and ARV knowledge on viral suppression" is not methodologically adequate given the unmeasured adherence level among participants, different samples sizes used for knowledge assessment and viral suppression, and the fact that it's not included as an objective of this study. I, therefore, suggest that the portion be expunged from the paper.

Reviewer #2: The content of the manuscript is scientifically relevant and the findings are in line with similar papers coming out of this region. There are certain issues which I would like to point out..

1) The language has to be considerably improved throughout the manuscript. I would suggest seeking help from an English language editor.

2) Sample size calculation and sampling strategy is not mentioned in the manuscript. Was the sample universal and only those fulfilling the inclusion criteria were included?

3) The scoring assigned for the knowledge section of the questionnaire is not clearly explained and is quite confusing.

4) What is the difference between does not know and do not answer ---is the difference literal?

5) I would suggest retaining the correct responses and arrange it in descending order.

6) Some of the statements under Knowledge regarding ARV are assessing perception, rather than knowledge

7)Class interval chosen for age groups are not uniform? what is the basis for such interval?

8) For tables determining the factors for not achieving viral suppression, the information is non-coherent with loads of variables. The interpretation becomes too difficult.

6. PLOS authors have the option to publish the peer review history of their article (what does this mean?). If published, this will include your full peer review and any attached files.

Reviewer #1: **Yes: **Akinola Ayoola Fatiregun

Reviewer #2: No

---

## [Author Response · Author response to Decision Letter 0]

4 Jul 2021

To whom it may concern

Thank you very much for your comments. We have addressed each one of your comments/recommendations in the rebuttal letter.

Best regards

Jose Luis Paredes

---

## [Decision Letter · Decision Letter 1]

4 Aug 2021

HIV and antiretroviral treatment knowledge gaps and psychosocial burden among persons living with HIV linked to care in Lima, Peru

PONE-D-21-01332R1

Dear Dr. Paredes,

We’re pleased to inform you that your manuscript has been judged scientifically suitable for publication and will be formally accepted for publication once it meets all outstanding technical requirements.

Kind regards,

Rekha Thapar, MD

Guest Editor

PLOS ONE

Additional Editor Comments (optional):

Reviewers' comments:

Reviewer's Responses to Questions

**Comments to the Author**

1. If the authors have adequately addressed your comments raised in a previous round of review and you feel that this manuscript is now acceptable for publication, you may indicate that here to bypass the “Comments to the Author” section, enter your conflict of interest statement in the “Confidential to Editor” section, and submit your "Accept" recommendation.

Reviewer #1: All comments have been addressed

Reviewer #3: All comments have been addressed

2. Is the manuscript technically sound, and do the data support the conclusions?

Reviewer #1: Yes

Reviewer #3: Yes

3. Has the statistical analysis been performed appropriately and rigorously? 

Reviewer #1: Yes

Reviewer #3: Yes

4. Have the authors made all data underlying the findings in their manuscript fully available?

Reviewer #1: Yes

Reviewer #3: Yes

5. Is the manuscript presented in an intelligible fashion and written in standard English?

Reviewer #1: No

Reviewer #3: Yes

6. Review Comments to the Author

Reviewer #1: All previous comments were addressed appropriately. However, the tables will need to be formatted or design in line with the journal's guideline. In addition, definitions of all acronyms used within the table should be provided as table footnotes or legend.

Reviewer #3: Some of my observations and queries:

1) The manuscript needs language edit

2) Authors have failed to mention 171 eligible participants were selected from from what sample ? How many PLHIV were there in the centre during the study period ?

3) Why was the sample size calculation not done ?

4) The knowledge section in the study was a mixture of both knowledge and awareness about HIV / ARV? On what basis was answering 80% of the questions correctly was considered good knowledge?

5) Association between viral suppression and good knowledge is not discussed in the paper.

7. PLOS authors have the option to publish the peer review history of their article (what does this mean?). If published, this will include your full peer review and any attached files.

Reviewer #1: **Yes: **Dr Akinola Ayoola Fatiregun

Reviewer #3: No

---

## [Editor Report · Acceptance letter]

11 Aug 2021

PONE-D-21-01332R1 

HIV and antiretroviral treatment knowledge gaps and psychosocial burden among persons living with HIV in Lima, Peru 

Dear Dr. Paredes:

I'm pleased to inform you that your manuscript has been deemed suitable for publication in PLOS ONE. Congratulations! Your manuscript is now with our production department. 

Kind regards, 

on behalf of

Dr. Rekha Thapar 

Guest Editor

PLOS ONE